# Thermal disruption of a Luttinger liquid

Danyel Cavazos-Cavazos [1], Ruwan Senaratne[1], Aashish Kafle [1] & Randall G. Hulet [1] ✉

The Tomonaga–Luttinger liquid (TLL) theory describes the low-energy excitations of strongly correlated one-dimensional (1D) fermions. In the past years, a number of studies have provided a detailed understanding of this universality class. More recently, theoretical investigations that go beyond the standard low-temperature, linear-response TLL regime have been developed. While these provide a basis for understanding the dynamics of the spin-incoherent Luttinger liquid, there are few experimental investigations in this regime. Here we report the observation of a thermally induced, spin-incoherent Luttinger liquid in a $^6$Li atomic Fermi gas confined to 1D. We use Bragg spectroscopy to measure the suppression of spin-charge separation and the decay of correlations as the temperature is increased. Our results probe the crossover between the coherent and incoherent regimes of the Luttinger liquid and elucidate the roles of the charge and the spin degrees of freedom in this regime.

Studies of strongly interacting atomic gases in 1D, aided by exactly solvable models[1–6], have provided remarkable insight into the physics of highly correlated quantum many-body systems in regimes that are increasingly accessible to experiment[7–14]. The low-energy properties of spin-1/2 fermions in 1D are well understood in terms of the TLL theory[15–19], which features low-energy collective spin- and charge-density waves (SDWs/CDWs). These sound waves propagate with different velocities, thus resulting in a spin-charge separation. At its core, the TLL universality class is characterized by collective excitations that are coherent and linearly dispersing. Several regimes, however, have been found to extend beyond this spin-charge separation paradigm, allowing access to new classes of unconventional Luttinger liquids where the coherence of the excitations is disrupted[20–22]. Higher-order effects such as band-curvature and back-scattering, for example, produce a nonlinear Luttinger liquid[23], for which the linearity of the dispersion is disrupted. Spin polarization is expected to control a quantum phase transition, at which the TLL turns quantum critical and all thermodynamic quantities exhibit universal scaling[22]. Allowing for anisotropic coupling between 1D chains of fermions could realize the sliding Luttinger liquid (SLL) phase[24]. Topological materials such as single- and bi-layer graphene[25] provide access to phases such as chiral Luttinger liquids[26] ($\chi$LL), which host excitation modes with a preferred sense of propagation.

Finite temperature represents another pathway for disrupting the correlations in a TLL (Fig. 1a). In the low temperature ($T$) limit, the thermal energy $k_B T$ is the lowest energy scale, and both the charge- and spin-density waves propagate coherently in accordance with the standard TLL theory, thus defining the spin-coherent (SC) regime. As $T$ is increased, thermal fluctuations disrupt the coherence in the spin sector first, and the system enters the spin-incoherent (SI) Luttinger liquid regime[27]. In the SI regime, spin-spin correlations are expected to exhibit a rapid exponential decay, while the density–density correlations retain a slower algebraic decay, leading to correlations that are independent of the spin sector[28]. The SI regime has been investigated theoretically with the Bethe ansatz[28–30] and a bosonized path integral approach[31–33] to describe both fermions[34] and bosons[35]. Recent studies have also identified density correlations[28,34,36] that distinguish the SC and the SI regimes. Experimental evidence for the SI regime, however, remains scarce. Studies of quasi-1D solid-state materials using angle-resolved photoemission spectroscopy[37,38] have suggested that signatures of the SI regime arise for small electron densities[39]. The control and tunability afforded by ultracold gases, on the other hand, facilitate the systematic study of Luttinger liquid physics[10–12,14].

Here, we explore the crossover between a SC Luttinger liquid and the SI regime in a pseudo-spin-1/2 gas of $^6$Li atoms loaded into an array of 1D waveguides. We use Bragg spectroscopy to show the suppression of spin-charge separation and the systematic loss of coherence with increasing $T$. Surprisingly, signatures of the spin degree of freedom persist even for $T > T_F$, where $T_F$ is the Fermi temperature.

[1]Department of Physics and Astronomy, Rice University, Houston, Texas 77005, USA. ✉e-mail: randy@rice.edu

## Results

### Accessing the spin-incoherent regime in a TLL

Spin-charge separation results in a separation of energy scales for the spin and the charge sectors of the TLL Hamiltonian. These are given by $E_{s,c} = \hbar n v_{s,c}$, where $v_s$ and $v_c$ are the propagation speeds of each mode, and $n$ is the 1D density[27]. The speed of the SDW is less than that of the CDW[1], and thus $E_s < E_c$ in the SC regime. In the SI regime, where $E_s < k_B T < E_c$, the spin configurations are mixed, even though the charge correlations remain unaffected. Consequently, it is expected that the SDW no longer propagates in the SI regime, whereas the CDW continues to propagate[33]. For sufficiently high $T$, such that $k_B T > E_c$, the coherence in both sectors is expected to vanish, corresponding to the thermal regime. The interplay between $T$, interaction strength, and waveguide occupancy $N$ defines an energy hierarchy for the Luttinger liquid, as shown schematically in Fig. 1b. These regimes are expected to be separated by smooth crossovers, and the transition between them remains a subject of active research[27].

Our methods for preparing and probing quantum degenerate, pseudo-spin-1/2 Fermi gases of $^6$Li atoms and characterizing them by Bragg spectroscopy[12,14,40] are described in "Methods." The pseudo-spin-1/2 system consists of a balanced spin mixture of the lowest and third-to-lowest hyperfine ground states of $^6$Li, which we label as $|1\rangle$ and $|3\rangle$. The interactions depend on the s-wave scattering length, $a$, which is fixed to be 500 $a_0$, where $a_0$ is the Bohr radius, by using the $|1\rangle$-$|3\rangle$ magnetic Feshbach resonance located at 690 G[41]. We found that 500 $a_0$ s the largest value of $a$ achievable without incurring an unacceptably large atom loss arising from 3-body recombination[12]. We vary $T$ by modifying the duration and depth of the evaporative cooling

trajectory first in a crossed-beam dipole trap and then in a 3D harmonic trap. Following evaporation, the atoms are loaded into a 3D optical lattice and then into a 2D optical lattice with a depth of 15 $E_r$, where $E_r = k_B \times 1.4$ µK is the recoil energy due to a lattice photon of wavelength 1064 nm. The result is a sample of $6.5 \times 10^4$ atoms distributed over an array of quasi-1D tubes.

The Gaussian curvature in our confining beams results in an inhomogeneous number profile, $N(r)$, where $r$ is the cylindrical coordinate perpendicular to the axis of each tube. We partially compensate for this effect by introducing anti-confining, single-passed laser beams (532 nm) along each of the three orthogonal directions during the 3D lattice ramping[12,40]. Adjusting the power of the anti-confining beams allows us to maintain a comparable $N(r)$ profile for each value of $T$ within a range $\Delta T \lesssim 1$ µK. We focused our studies on the range of 500–1500 nK, which is found to encompass the SI and thermal regimes while still distinguishing a clear spin-charge separation ($v_s < v_c$) at the lower end of the range of $T$. Our lowest $T$ of 500 nK is approximately twice the temperature used in ref. 14. Because the highest $T$ accessed is well below the radial confinement energy, the atoms are in the quasi-1D regime in all cases.

Bragg spectroscopy can be used to separately excite density waves in the charge or spin sectors by appropriately choosing the detuning, $\Delta$, of the Bragg beams with respect to a virtual excited state (Fig. 2). For $\Delta > 0$, blue-detuned light, an atom will be attracted to regions of low light intensity. Conversely, for $\Delta < 0$, red-detuned light, an atom will be attracted to regions of high light intensity. By controlling the magnitude and sign of $\Delta$ for each spin state, we create a light shift that is effectively equal in magnitude and sign for both spins ($\Delta_\sigma = 11$ GHz) or one that is equal in magnitude but opposite in sign ($\Delta_\alpha = \pm 80$ MHz). We refer to the former as a symmetric Bragg

**Fig. 1 | Energy hierarchy of a Luttinger liquid. a** Schematic diagram showing the energy regimes of a Luttinger liquid in the spin-coherent (SC), spin-incoherent (SI), and thermal regimes, illustrating the effect of decoherence of the spin and charge correlations. **b** Crossover hierarchy of a quasi-1D atomic Fermi gas (see "Methods"). The incoherent regimes can be reached either by increasing the scattering length $a$, increasing the temperature $T$, or by reducing the number of atoms per tube, $N$. Dashed lines correspond to the boundaries between the different regimes, defined by $E_s$ and $E_c$, which are functions of $a$ and $N$. The solid line illustrates a trajectory corresponding to constant $N$ and $a$. As $T$ is increased, the system first loses its spin coherence for $E_s < k_B T < E_c$, and at a sufficiently high $T$, such that $E_s < E_c < k_B T$, all coherence in the system is lost.

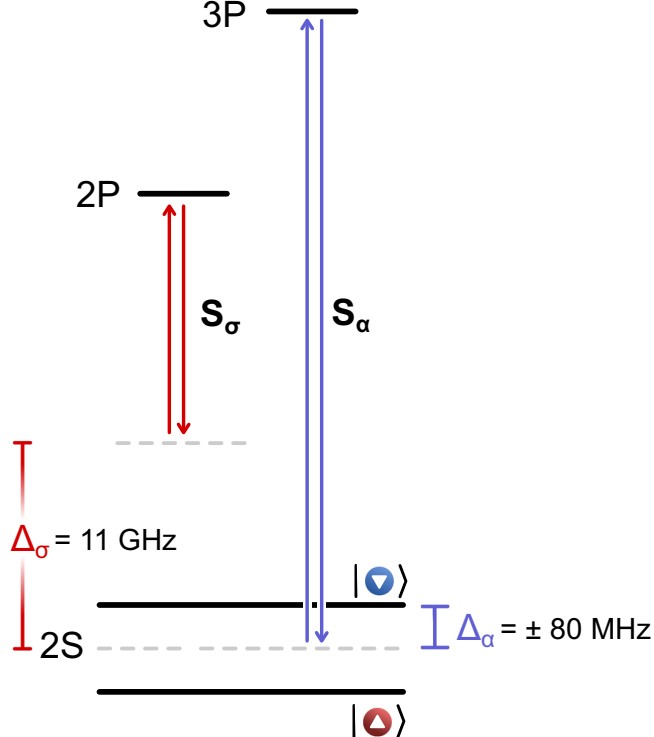

**Fig. 2 | Symmetric and antisymmetric excitations via Bragg spectroscopy.** Partial energy-level diagram of $^6$Li showing relevant transitions and laser detunings of the Bragg pulses. We generate a symmetric light shift by symmetrically detuning the frequency of the Bragg beams far ($\Delta_\sigma = 11$ GHz) from the 2S → 2P resonance. For an antisymmetric excitation, the Bragg beams are detuned by $\Delta_\alpha = \pm 80$ MHz from the 2S → 3P resonance frequencies.

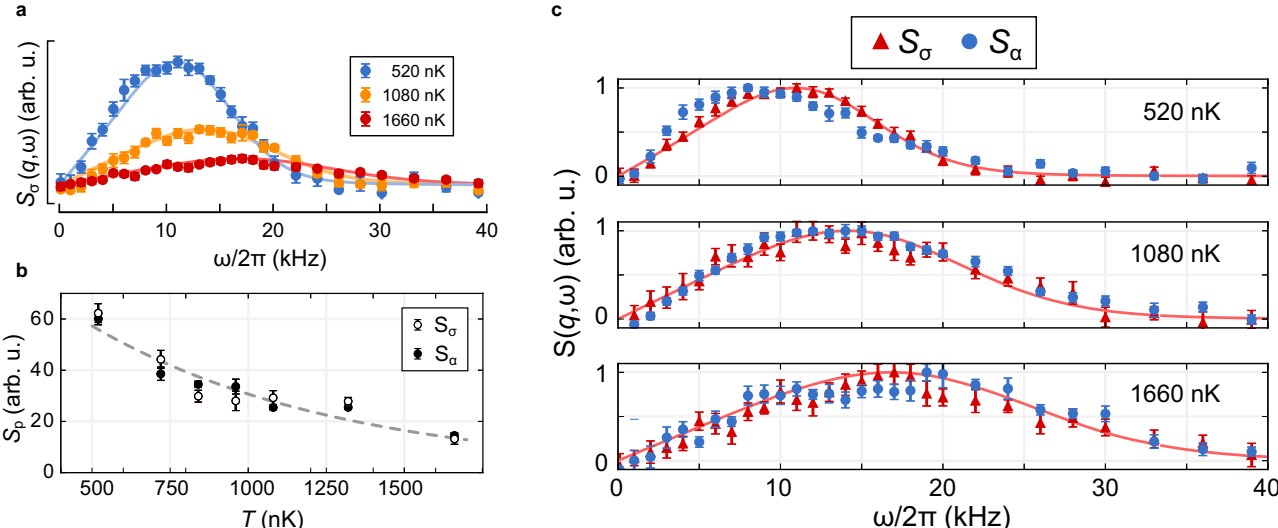

**Fig. 3 | Temperature effects on density waves. a** Bragg signal corresponding to symmetric excitation, $S_\sigma(q,\omega)$, for different $T$. Solid lines are a fit to the Bragg spectra using a free-fermion theory with fit parameter $T$ (see "Methods"). **b** Peak amplitude $S_p$ of the Bragg spectra as a function of $T$, where empty circles correspond to a symmetric Bragg excitation, $S_\sigma$, while filled circles represent the antisymmetric excitation, $S_\alpha$. The dark-gray dashed line is a fit to $T$, assuming an exponential dependence, signaling the loss of correlations due to thermal fluctuations. **c** Normalized Bragg signals corresponding to symmetric ($S_\sigma(q,\omega)$, red triangles) and antisymmetric ($S_\alpha(q,\omega)$, blue circles) excitations for different T. Symmetric data and associated fits (solid lines) are the same as in (**a**). Each data point in (**a**) and (**c**) is the average of at least 20 separate experimental shots. Error bars represent standard error obtained via bootstrapping (see "Methods" for details).

configuration and the latter as the asymmetric one, designated by subscripts $\sigma$ and $\alpha$, respectively. Since the Bragg pulse imparts both a momentum $\hbar q$ and an energy quantum $\hbar\omega$ per atom, we can determine the speed of propagation of the excitations (see "Methods"). The Bragg-induced momentum kick results in outcoupling a fraction of the atoms, the number of which is proportional to the total momentum delivered to the ensemble and constitutes the measurement signal[10,12,14]. For a spin-balanced sample, the symmetric Bragg pulse exclusively excites the CDW, and the measured signal is proportional to the dynamic structure factor (DSF) $S_c(q,\omega)$ of the gas[42–44], which encodes the density–density correlations[45] in the charge sector. Conversely, the antisymmetric Bragg pulse exclusively excites the SDW, and the measured signal is proportional to the DSF $S_s(q,\omega)$, which encodes the spin-density-spin-density correlations. We extract the most probable value of the propagation speed, $v_p$, from the measured peak frequency $\omega_p$ for each spectrum using the relation $v_p = \omega_p/q$.

**Suppression of spin-charge separation**

We have previously exploited this individual addressability of the CDW/SDW by using Bragg spectroscopy to characterize spin-charge separation in the SC regime by measuring $v_{c,s}$ as functions of repulsive interaction[14]. In this work, we demonstrate that in the SI regime where spin-order fluctuations are dominant, the assumption of local spin-balance is no longer justified and leads to the loss of individual addressability of the CDW/SDW using Bragg processes. To emphasize this notion, we label the measured signals as $S_\sigma$ and $S_\alpha$ for symmetric and antisymmetric light shifts, instead of $S_c$ and $S_s$, with corresponding propagation speeds $v_\sigma$ and $v_\alpha$.

Representative Bragg spectra, corresponding to several values of $T$, are shown in Fig. 3a. We determine $T$ by fitting the measured $S_\sigma(q,\omega)$ to a free-fermion theory for which the density inhomogeneity is accounted for by the local density approximation, as in our previous work[12,14]. We observe a suppression of the peak excitation amplitude for the symmetric configuration with an exponential dependence on $T$ (Fig. 3b), in agreement with a previous theoretical study of the role of temperature that predicted exponential decay of correlations with increasing $T$[28]. The decay in the Bragg response

arises from the loss of density–density correlations due to the proliferation of holes.

A standard prediction in the SI regime is that only the charge-mode can propagate coherently, which suggests that the gas will only respond to a symmetric Bragg configuration (charge mode)[29,33]. However, we observe that the measured peak amplitudes are the same for both configurations within uncertainty (Fig. 3b). While surprising, this may be because the antisymmetric Bragg configuration excites the CDW in the SI regime. During the crossover from the SC regime to the SI regime, the Luttinger liquid will be effectively averaged over an increasing number of spin configurations that have regions of local spin imbalance (see Fig. 1a). Because of these regions, induced by thermal fluctuations, the antisymmetric Bragg configuration no longer has a locally antisymmetric response. Rather, with increasing T, this Bragg pulse progressively couples to the charge-mode as the system crosses into the SI regime.

**Spin correlations in the SI regime**

Our measurements confirm that in the SI regime, the peak excitation frequency and the high-energy tails of $S(q,\omega)$ for each sector are identical within their uncertainty (Fig. 3c). This condition is clearly different from the SC regime, where in addition to the shift in $\omega_p$ due to the spin-charge separation, the spectra corresponding to SDW excitations have enhanced high-frequency tails that are related to nonlinear interaction effects that are exclusive to the spin sector[14]. The match between the symmetric and antisymmetric excitation spectra at sufficiently high temperatures is indicative of the charge-mode character of the excitation induced by the antisymmetric pulse.

This argument is validated by observing a gradual suppression of the difference between $v_\sigma$ and $v_\alpha$ (Fig. 4). In the SC regime, we can clearly distinguish the two fundamental types of density waves (CDWs and SDWs) due to their distinct propagation speeds. This distinction is lost in the SI regime and suggests a sole propagating mode, as expected theoretically[29,33]. We estimate an upper bound for the crossover temperatures by evaluating the thermal hierarchy at the center of a single tube located at the center of the ensemble, where the maximum occupancy is $N \simeq 30$. We extract the density at the center

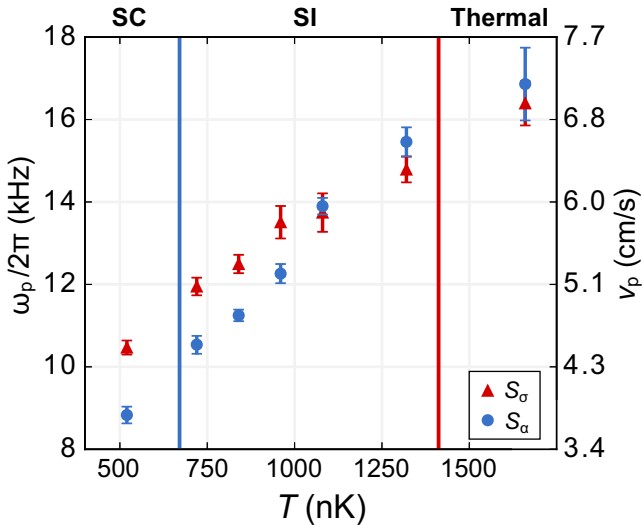

**Fig. 4 | Suppression of spin-charge separation.** Frequencies at the peak amplitude of measured Bragg spectra for symmetric (red triangles) and antisymmetric (blue circles) pulse configurations as a function of $T$. The corresponding speed of sound $v_p = \omega_p/q$ is given by the right axis. Error bars are the statistical standard errors of the extracted peak frequency obtained via a quadratic regression (see "Methods"). At sufficiently high $T$, a separation between the measured velocities is lost, consistent with the suppression of the spin-density wave. Solid vertical lines correspond to the boundaries of the thermal hierarchy evaluated for $N = 30$ and $a = 500\ a_0$.

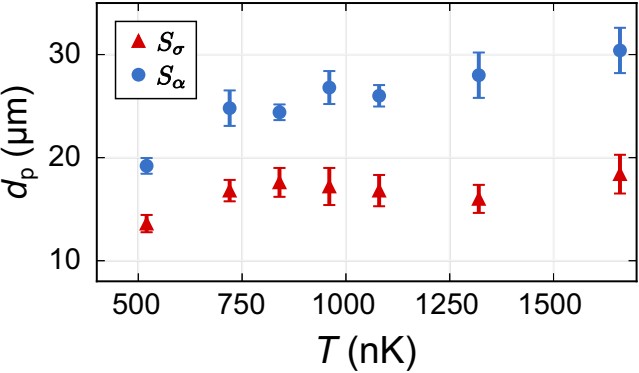

**Fig. 5 | Dispersion of incoherent density waves.** $1/e^2$ axial width of outcoupled atoms, $d_p$, following a Bragg pulse and 150 μs time-of-flight expansion for symmetric (red triangles) and antisymmetric (blue circles) excitations as a function of $T$. The widths are the Gaussian fits to the positive outcoupled signal at $\omega_p$[14]. Error bars are standard errors determined by bootstrapping for at least 20 independent images. The spin-density waves show an increased width arising from the reduced lifetime of the SDW excitation caused by back-scattering. The difference between the symmetric $S_\sigma(q, \omega)$ and antisymmetric $S_\alpha(q, \omega)$ excitations remains, even after the system is fully thermal.

of each tube (and consequently the Fermi velocity $v_F$) from in situ phase-contrast images of the atom cloud[12,14] (see "Methods"). The propagation speeds for either mode at zero temperature can then be obtained from a Bethe ansatz calculation[46], and results in the energy scales $E_s \sim k_B \times 630$ nK and $E_c \sim k_B \times 1330$ nK, which are shown as solid lines in Fig. 4. For $T > 1000$ nK, we find that $v_\sigma \simeq v_\alpha$, to within our uncertainty, which is consistent with a sole propagating mode. Our results provide an experimental demonstration of a thermal disruption of spin-charge separation.

We further characterized the Bragg spectra as a function of $T$ by measuring the width $d_p$ of the packet of atoms outcoupled by the Bragg pulse after time-of-flight expansion (Fig. 5). This width is a probe of the nonlinearity of the dispersion of the excitation; a larger width indicates increased nonlinearity. We have previously shown that in the SC regime with strong repulsive interactions, $d_p$ for the SDW is larger than for the CDW, indicating nonlinear effects in the spin-mode dispersion that are not present in charge excitations. We found that the nonlinearity is related to the finite lifetime of the SDW excitation due to back-scattering[14]. In the SI regime, although the symmetric and antisymmetric Bragg spectra are indistinguishable for sufficiently high temperatures, we nonetheless observe a difference in $d_p$ between the two Bragg configurations, even after the system has become fully thermal. This is perhaps surprising given the expectation of universal spin physics in the SI regime and suggests further study of the effects of nonlinearity on the decay of spin correlations is necessary[28].

## Discussion
In this manuscript, we have characterized the temperature crossover between a coherent and an incoherent Luttinger liquid, as evidenced by the exponential decay of correlations as the system transitions into the SI regime and the full suppression of spin-charge separation —a hallmark of the TLL theory—clearly demonstrating the signatures of a disrupted Luttinger liquid. Further work using spin-sensitive imaging can focus on the measurement of density–density

correlation functions, as well as exploring the anomalous exponents in the decay of charge correlations[28]. Our measurements can also be readily extended to a systematic study of more exotic regimes of 1D fermions. A gas with attractive interactions ($a < 0$) is predicted to realize the Luther–Emery liquid phase[47], which exhibits a gap only in the spin sector and is a potential 1D analog of a superconductor[48,49]. Another interesting path is the characterization of a spin-imbalanced sample, where a quantum-critical region is expected to appear at finite temperatures for repulsive interactions[22], and new emergent liquid and gas-like quantum phases near a quantum phase transition could potentially be studied.

## Methods
### Temperature control for different samples
The final temperature $T$ is determined by the evaporative cooling trajectory realized in a 1070-nm crossed-beam dipole trap, followed by further evaporation in a 3D harmonic trap produced by the intersection of three mutually orthogonal focused trapping beams of wavelength 1064 nm. We vary the duration and final evaporation depth at each stage to control $T$ while maintaining an approximately constant total atom number. The temperature has a significant effect on the tube-to-tube number distribution $N(r)$. The central tube occupancy is highest for low $T$, and it diminishes with increasing $T$. We partially compensate for these variations by introducing repulsive (532 nm) compensation laser beams along the three lattice directions, which are ramped on during the turn-on of the 3D optical lattice[40]. This enables us to adjust the degree of confinement in the optical lattice so that $N(r)$ is made approximately independent of $T$. The spin-charge separation we report in Fig. 4 for the lowest $T$ is smaller in magnitude than in our previous report[14]. This difference is a consequence of the lowest $T$ being 500 nK in the present experiment, while $T = 250$ nK for the experiment reported in ref. 14.

### Two-photon Bragg spectroscopy
The experimental conditions for probing and analyzing the resulting low-energy excitation spectra are the same as those previously reported in ref. 14. The Bragg pulse duration is 200 μs, and the atoms are imaged after 150 μs of time-of-flight. We adjust the angle between the Bragg beams such that for both modes, the Bragg wave-vector is parallel to the tube axis and has a magnitude $|\mathbf{q}| = 1.47\ \mu m^{-1}$, corresponding to $0.3\ k_F$ for the central tube. We calculate the Bragg signal by

quantifying the number of atoms that receive a momentum kick from the Bragg pulse as a function of $\omega$. This Bragg signal is proportional to the dynamic structure factor $S(q,\omega)$[10,12,14,43,44].

## Crossover hierarchy evaluation

The energy scales for the charge and spin sectors can be approximated as $E_\eta = \hbar n v_\eta$ (ref. [27]), where $n$ is the 1D density, $v_\eta$ is the propagation velocity of each mode and $\eta$ = s,c correspond to either charge or spin sectors, respectively. We express $E_\eta$ as $E_\eta(a,N)$, with $a$ being the s-wave scattering length and $N$ the tube occupancy. The density $n(a,N)$ is calculated by using the local density approximation (LDA)[12,14], where we numerically solve the equation:

$$\mu - \frac{1}{2}m\omega_z^2 x^2 = \frac{\hbar^2\pi^2}{8m}[n(x)]^2 + \frac{g(a)}{2}[n(x)], \quad (1)$$

where $\mu$ is the chemical potential defined by $N = \int n(x)dx$, $m$ is the atomic mass, $\omega_z$ is the axial angular trapping frequency, and $g(a) = \frac{2\hbar^2}{m}\left(\frac{a}{a_\perp^2}\right)\frac{1}{1-C(a/a_\perp)}$ is the interaction strength[50], where $C = |\xi(1/2)|/\sqrt{2} \sim 1.03$ and $a_\perp = \sqrt{\hbar/m\omega_\perp}$ is the length scale of the transverse harmonic confinement for a radial angular trapping frequency $\omega_\perp$. The propagation velocity for each mode $v_{s,c}$ can be expressed as $v_\eta = v_F\beta_\eta$, where $v_F$ is the Fermi velocity, explicitly given by $v_F = \sqrt{\frac{\hbar}{m}N\omega_z}$. The factor $\beta_\eta$ can be calculated exactly from the zero-temperature Bethe ansatz[46], although a first-order approximation can be given as

$$\beta_\eta \simeq \sqrt{1 \pm \frac{2\gamma}{\pi^2}}, \quad (2)$$

where the "+" sign corresponds to charge and the "−" sign to spin, and $\gamma$ is the Lieb–Liniger parameter:

$$\gamma(a,n(a,N)) = \frac{a}{a_\perp^2 n}\frac{1}{1-C(a/a_\perp)}. \quad (3)$$

Thus, we can express the energy scales as:

$$E_\eta(a,N) \simeq \hbar n(a,N)v_F(N)\sqrt{1 \pm \frac{2\gamma(a,n(a,N))}{\pi^2}}. \quad (4)$$

For Fig. 1, we evaluated the density at the center of a waveguide characterized by $\omega_\perp = 2\pi \times 227$ kHz and $\omega_z = 2\pi \times 1.3$ kHz, which corresponds to the quasi-1D geometry created by a 2D optical lattice with a depth of 15 $E_r$. The energy hierarchy defines the spin-coherent (SC), spin-incoherent (SI), and thermal regimes:

$$\textbf{SC} \quad k_B T < E_s < E_c \quad (5)$$

$$\textbf{SI} \quad E_s < k_B T < E_c \quad (6)$$

$$\textbf{Thermal} \quad E_s < E_c < k_B T. \quad (7)$$

We evaluate the boundaries for the SC-SI and SI-thermal regimes, shown with dashed lines in Fig. 1b, by the conditions $E_s = k_B T$ and $E_c = k_B T$, respectively. For our parameters, $E_s \sim k_B \times 630$ nK and $E_c \sim k_B \times 1330$ nK.

## Data availability

All data needed to evaluate the conclusions in the paper are archived on Zenodo[51]. All other data that support the plots within this paper and other findings of this study are available from the corresponding author upon request.

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

## Acknowledgements

We thank G. Fiete and H. Pu for their helpful suggestions. This work was supported in part by the Army Research Office Multidisciplinary University Research Initiative (Grant No. W911NF-17-1- 0323) and the NSF (Grant No. PHY-2011829). D.C.-C. acknowledges financial support from CONACyT (Mexico, Scholarship No. 472271).

## Author contributions

D.C.-C., R.S., and A.K. performed the measurements and analyzed data. All work was supervised by R.G.H. All authors discussed the results and contributed to the manuscript.

## Competing interests

The authors declare no competing interests.
