## [Peer Review File · Nature Communications]

Thermal disruption of a Luttinger liquidREVIEWER COMMENTS

Reviewer #1 (Remarks to the Author):

The authors study experimentally the dynamics of spin-1/2 fermions confined to one spatial dimension. The system has two intrinsic energy scales, associated with propagation of spin and charge excitations. These scales are widely separated parametrically when the interparticle interactions are strong: charge excitation energy should be compared with the Fermi energy E_F , and spin excitation energy should be compared with E_F/γ , where γ is a dimensionless interparticle interaction strength. The system stays in the quantum coherent regime (called the spin coherent, SC, in the manuscript) when its temperature $T < E_F/\gamma$. Temperature fluctuations are dominant in the regime $T > E_F/\gamma$, which is called the charge incoherent, CI, in the manuscript. The intermediate regime $E_F/\gamma < T < E_F$ is called the spin incoherent, SI. The manuscript reports the experimental realization of the SI regime.

The experimental setup used to reveal the SI regime has been implemented earlier by the same group, most recently in Ref. [14]. However, the parameter range used in this manuscript has not been accessed before. In particular, the temperature of the 1D gas ranges from 500 to about 2000 nK, while it does not go above 250nK in Ref. [14]. This way, the manuscript presents the studies in the yet unexplored regime, justifying the novelty of the subject.

The manuscript demonstrates that spin and charge excitations are coupled to each other in the SI and CI regimes. More specifically, the velocities of sound for spin and charge excitations take very different values when the interparticle repulsion is large, see, for example, Fig. 3 in Ref. [45]. The authors of the manuscript excite the system by the two consecutive Bragg pulses in such a way that the collective spin and charge density waves should emerge, would the system be in the spin-charge separated regime, described by the Luttinger Liquid low-energy effective field theory. However, the authors observe two collective excitations propagating with nearly the same velocities, in clear contrast with what would have followed from the spin-charge separated dynamics.

Exact analytic results for the dynamical correlation functions in the SI and CI regimes for the large but finite interaction strength and finite temperature are not present in the literature so far. I believe that this experimental work will boost the theoretical research towards obtaining them, as well as further experimental studies of the SI and CI regimes.

Overall, I find the present manuscript very exciting, and recommend it for the publication. I have several minor suggestions, which the authors may address in the revised manuscript. As a general advice, I would suggest that the authors reread the manuscript standing from the point of view of a general reader not fully familiar with the subject:

1. page 3 line 1 "Consequently, the CDW remains the dominant

propagating mode[32]" -- later the authors demonstrate that in their experiment CDW and SDW are mixed. Hence, the above statement is not a general statement about the dynamics in the SI regime. I believe I understand the source of that statement, but unprepared readers may be disoriented.

2. page 3 line 5: "properties are not well understood [27]" -- Ref. [27] is a (short) review paper. It is not aimed to demonstrate that the SI regime is not well understood, hence the reader should invent the reasons alone. Sounds discouraging.

3. page 3 line 23: " $v_s < v_c$ " -- neither v_s nor v_c have not been introduced at this point.

4. page 4 line 8: " $S_c(q, \omega)$ " -- Fig. 2 is discussed around that expression. There is no S_c in the figure and in the caption. The difference between S_c and S_σ has not been explained at this point.

Reviewer #2 (Remarks to the Author):

The authors report on the experimental study of Li atoms in quasi-one-dimensional tubes. They use Bragg spectroscopy to excite density waves in the charge and spin sectors. The authors present a clear qualitative introduction and description of the experiment. However, their interpretation of the experimental results raises a lot of questions.

The primary tool of the research is based on the following statement: "Bragg spectroscopy can be used to separately excite density waves in the charge (spin) sector by appropriately detuning the Bragg beams to generate a symmetric (antisymmetric) light shift..." The authors then explain that in the spin-incoherent (SI) regime, "The atoms should respond only to a Bragg excitation with a symmetric light shift in the SI regime, since spin correlations are suppressed there". Therefore, it is unclear why the authors suggest labeling the measured signals as symmetric and antisymmetric, instead of charge and spin. Also, the sentence about the Bragg pulse used to excite spin density wave now couples to the charge mode is confusing. Does it mean that in SI regime, both symmetric and antisymmetric pulses excite only charge density wave?

The Bragg signal for a symmetric light shift is presented in Fig. 2a for different temperatures, and equal peak amplitudes for symmetric and antisymmetric excitations are shown in Fig. 2b. Are Bragg curves identical for both signals and only shifted by the frequency at low temperatures as shown in extended Fig. 2? Fig. 2a is friendly and easy to understand – the charge density wave is clearly observed in the SI regime and is suppressed at high temperatures, which authors call a charge-incoherent regime. Then one would naively expect that if the spin density is suppressed already in the SI regime, then it must be seen from the antisymmetric Bragg signal.

Do different wave velocities in the SI regime in Fig. 3 mean that there are two waves?

Why different widths of the outcoupled atoms in Fig. 4 are the consequences of spins? More, earlier, the authors write that the size of a fraction of the outcoupling atoms is proportional to the dynamic structure factor and present Bragg signals based on this statement. Then the widths are different at high temperatures, whereas sizes are equal. What is the difference between the width and the size? To summarize, the manuscript presents a few interesting results but lacks clear discussions and needs more experimental data. In my opinion, it does not correspond to the high standards of Nature Communications, and I cannot recommend its acceptance.

We thank both reviewers for their time and effort to provide thoughtful comments and questions, and for their generally positive evaluations. We have thoroughly addressed these comments, as described below. As a result, we have made substantive changes to the manuscript, including the addition of a new figure (Fig. 2) introducing Bragg spectroscopy, and a new subpanel in Fig. 3 showing data that we believe are relevant to the Reviewers' concerns. Additionally, we have changed the title to be more descriptive: "Thermal Disruption of a Luttinger Liquid". We believe that this title more accurately describes the main finding of the work.

In what follows, we address, one by one, each of the reviewer comments and questions. Blue text denotes the reviewer, while our responses are given in black. We also submit two versions of the revised text, one of which indicates major deletions in red and additions in blue. The line numbers referred to below pertain to the final version.

Reviewer: 1

The authors study experimentally the dynamics of spin-1/2 fermions confined to one spatial dimension. The system has two intrinsic energy scales, associated with propagation of spin and charge excitations. These scales are widely separated parametrically when the interparticle interactions are strong: charge excitation energy should be compared with the Fermi energy E_F , and spin excitation energy should be compared with E_F/γ , where γ is a dimensionless interparticle interaction strength. The system stays in the quantum coherent regime (called the spin coherent, SC, in the manuscript) when its temperature $T < E_F/\gamma$. Temperature fluctuations are dominant in the regime $T > E_F$, which is called the charge incoherent, CI, in the manuscript. The intermediate regime $E_F/\gamma < T < E_F$ is called the spin incoherent, SI. The manuscript reports the experimental realization of the SI regime.

The experimental setup used to reveal the SI regime has been implemented earlier by the same group, most recently in Ref. [14]. However, the parameter range used in this manuscript has not been accessed before. In particular, the temperature of the 1D gas ranges from 500 to about 2000 nK, while it does not go above 250nK in Ref. [14]. This way, the manuscript presents the studies in the yet unexplored regime, justifying the novelty of the subject.

The manuscript demonstrates that spin and charge excitations are coupled to each other in the SI and CI regimes. More specifically, the velocities of sound for spin and charge excitations take very different values when the interparticle repulsion is large, see, for example, Fig. 3 in Ref. [45]. The authors of the manuscript excite the system by the two consecutive Bragg pulses in such a way that the collective spin and charge density waves should emerge, would the system be in the spin-charge separated regime, described by the Luttinger Liquid low-energy effective field theory. However, the authors observe two collective excitations propagating with nearly the same velocities, in clear contrast with what would have followed from the spin-charge separated dynamics.

Exact analytic results for the dynamical correlation functions in the SI and CI regimes for the large but finite interaction strength and finite temperature are not present in the literature so far. I believe that this experimental work will boost the theoretical research towards obtaining them, as well as further experimental studies of the SI and CI regimes.

Overall, I find the present manuscript very exciting, and recommend it for the publication. I have several minor suggestions, which the authors may address in the revised manuscript. As a general advice, I would suggest that the authors reread the manuscript standing from the

point of view of a general reader not fully familiar with the subject:

We thank Reviewer 1 for their positive remarks about our research, and for providing valuable feedback to improve the clarity of the manuscript. Many of the extensive changes made to the manuscript target the goal of clarifying the presentation. The purpose of the new Fig. 2, for example, is to better explain Bragg spectroscopy.

Comments and questions:

1) page 3 line 1 "Consequently, the CDW remains the dominant propagating mode[32]" -- later the authors demonstrate that in their experiment CDW and SDW are mixed. Hence, the above statement is not a general statement about the dynamics in the SI regime. I believe I understand the source of that statement, but unprepared readers may be disoriented.

Response: We have edited this phrase to read "Consequently, it is expected that the SDW no longer propagates in the SI regime, whereas the CDW continues to propagate", in order to emphasize that we are discussing the theoretical prediction of dynamics in the spin-incoherent regime (page 2, line 25). Additionally, in order to be consistent with existing literature we now refer to the charge-incoherent regime as the thermal regime.

2) page 3 line 5: "properties are not well understood [27]" -- Ref. [27] is a (short) review paper. It is not aimed to demonstrate that the SI regime is not well understood, hence the reader should invent the reasons alone. Sounds discouraging.

Response: We have edited this phrasing. The text now reads "... and the transition between them remains a subject of active research (Ref. 27)" (page 3, line 2).

3) page 3 line 23: " $v_s < v_c$ " -- neither v_s nor v_c have not been introduced at this point.

Response: We thank the reviewer for pointing this out. We have now introduced v_c and v_s in a previous sentence to avoid this confusion (page 2, line 22).

4) page 4 line 8: " $S_c(q, \omega)$ " -- Fig. 2 is discussed around that expression. There is no S_c in the figure and in the caption. The difference between S_c and S_σ has not been explained at this point.

Response: We again thank the reviewer for pointing this out. We now introduce S_σ and S_α earlier (page 4, line 17).

Reviewer: 2

The authors report on the experimental study of Li atoms in quasi-one-dimensional tubes. They use Bragg spectroscopy to excite density waves in the charge and spin sectors. The authors present a clear qualitative introduction and description of the experiment. However, their interpretation of the experimental results raises a lot of questions.

We thank Reviewer 2 for raising many important questions about our manuscript. We have substantially modified the discussion section to address the important concerns that were noted. We will now describe the modifications one by one.

1)The primary tool of the research is based on the following statement: “Bragg spectroscopy can be used to separately excite density waves in the charge (spin) sector by appropriately detuning the Bragg beams to generate a symmetric (antisymmetric) light shift...” The authors then explain that in the spin-incoherent (SI) regime, “The atoms should respond only to a Bragg excitation with a symmetric light shift in the SI regime, since spin correlations are suppressed there”. Therefore, it is unclear why the authors suggest labeling the measured signals as symmetric and antisymmetric, instead of charge and spin.

Response: We have expanded the discussion of the Bragg spectroscopy technique and have moved the figure illustrating the choice of detuning corresponding to each excitation configuration to the main text (now Fig. 2). We also explicitly discuss how Bragg spectroscopy allows for individual addressability of the CDW and SDW in the SC regime (page 4, line 13). However, it is not known *a priori* whether this remains true at elevated temperatures, which is why we instead choose to label the relevant measurements with subscripts alpha and sigma rather than spin and charge (added text, page 4, line 15).

2)Also, the sentence about the Bragg pulse used to excite spin density wave now couples to the charge mode is confusing. Does it mean that in SI regime, both symmetric and antisymmetric pulses excite only charge density wave?

Response: Our results are consistent with the antisymmetric Bragg configuration exciting the CDW in the SI regime. This is supported by the antisymmetric and symmetric excitation spectra being indistinguishable above a certain temperature (now shown in Fig. 3c), and by the theoretical expectation that only the CDW propagates in this regime. We suggest that this is due to spin-order fluctuations leading to local spin-imbalance, which leads to coupling of the antisymmetric pulse to the CDW. We have now made this explicit in our discussion (page 5, line 1).

3) The Bragg signal for a symmetric light shift is presented in Fig. 2a for different temperatures, and equal peak amplitudes for symmetric and antisymmetric excitations are shown in Fig. 2b. Are Bragg curves identical for both signals and only shifted by the frequency at low temperatures as shown in extended Fig. 2?

Response: In our previous work (Ref. 14) we measured the dynamical structure factor for the charge and spin modes to be identical only in the non-interacting case. At low T and for finite repulsive interaction strength, the spectra differ both in terms of the value of their peak frequencies, as well as in their shape. The antisymmetric Bragg spectrum (corresponding to the spin-mode at low T) has an enhanced high-frequency tail which we determined is due to backscattering collisions which introduce nonlinearity into the spin sector. In this work, we show

that at high T , the symmetric and antisymmetric Bragg spectra are identical both in peak frequency and shape, indicating that both Bragg configurations are exciting the same mode. We have added data that shows this in Fig. 3c and have added text discussing these matters (page 5, line 8).

4) Fig. 2a is friendly and easy to understand – the charge density wave is clearly observed in the SI regime and is suppressed at high temperatures, which authors call a charge-incoherent regime. Then one would naively expect that if the spin density is suppressed already in the SI regime, then it must be seen from the antisymmetric Bragg signal.

Response: We agree with the Reviewer that we expect the spin dynamic structure factor to be suppressed in the SI regime. However, we did not observe any variation in the ratio of the amplitudes of S_α to S_σ for any value of T (as shown in Fig. 3b). We interpret this result as evidence that the antisymmetric Bragg pulse is coupling to the charge-mode in the spin-incoherent regime. We attribute this charge character to the regions of local spin-imbalance that are induced in the sample due to thermal fluctuations. We have now revised the discussion to make this more explicit and to highlight how this is different from the standard prediction for the SI regime (page 4, line 27).

5) Do different wave velocities in the SI regime in Fig. 3 mean that there are two waves?

Response: At low T we can clearly distinguish the two fundamental types of density waves (CDWs and SDWs) in terms of their propagation speed. However, for sufficiently high temperatures, we cannot distinguish between the antisymmetric and symmetric excitation spectra, and that is consistent with the theoretical prediction of a sole propagating mode in the spin-incoherent regime (the CDW). We have now made this more explicit in our revised discussion (page 5, line 15). In the intermediate regime, we do observe a difference in peak frequency for the two Bragg configurations. This is expected, as the transition between the SC and SI regimes is expected to be a smooth crossover, and because our measurements are taken globally over an ensemble with density variation.

6) Why different widths of the outcoupled atoms in Fig. 4 are the consequences of spins? More, earlier, the authors write that the size of a fraction of the outcoupling atoms is proportional to the dynamic structure factor and present Bragg signals based on this statement. Then the widths are different at high temperatures, whereas sizes are equal. What is the difference between the width and the size?

Response: The size of the Bragg signal is proportional to the number of atoms that receive a Bragg kick, whereas the width d_p is the physical width of the kicked packet of atoms after a fixed amount of free expansion after release from the trap. The former is proportional to the total amount of momentum delivered to the sample, and, as the Reviewer points out, is the same for the two Bragg configurations. We now explicitly state that the Bragg signal is proportional to the number of kicked atoms (page 4, line 6).

The width d_p is proportional to the spread in momentum-space of atoms that are resonant with the Bragg pulse. In our previous work (Ref. 14) we determined that d_p is sensitive to the degree of nonlinearity in the TLL. In particular, we found that backscattering, a source of nonlinearity exclusive to the spin sector, causes an increase of d_p as a function of repulsive interaction in the spin sector. Here, we find that d_p is larger for the asymmetric configuration than the symmetric one for all temperatures at fixed repulsive interaction strength. This implies

that the asymmetric Bragg probe is sensitive to backscattering between distinct spins even in the thermal regime where there is neither spin nor charge coherence. We have expanded the discussion of d_p to include its relationship to the dispersive character of excitations (page 5, line 27).

To summarize, the manuscript presents a few interesting results but lacks clear discussions and needs more experimental data.

We agree with the Reviewer that our discussion of the results was unclear in the initial submission. We believe that our manuscript is significantly strengthened, however, as a result of the comments made by both Reviewers, and by the new data plot, now shown in Fig. 3c. We hope that the Reviewers and you now agree that our work is new, and that it provides quantitative insight into the propagation of density waves at finite T in 1D.

REVIEWERS' COMMENTS

Reviewer #2 (Remarks to the Author):

The authors have replied in detail to all my comments and have revised the manuscript accordingly. Now I can recommend its publication.

REVIEWERS' COMMENTS

Reviewer #2 (Remarks to the Author):

The authors have replied in detail to all my comments and have revised the manuscript accordingly. Now I can recommend its publication.

Response:

We thank the reviewers for providing useful and constructive comments to improve our manuscript.